# Taxonomic Revision of Genus *Ephedra* Tourn. ex L. in Egypt with Intra-Gender Diversity in Morphometric Traits and Fatty Acid Composition

**DOI:** 10.3390/plants13172442

**Published:** 2024-09-01

**Authors:** Maha H. Khalaf, Wafaa M. Amer, Najla A. Al Shaye, Mahmoud O. Hassan, Nasr H. Gomaa

**Affiliations:** 1Department of Botany and Microbiology, Faculty of Science, Beni-Suef University, Beni-Suef 62521, Egypt; dr_mody1983_science@yahoo.co.uk (M.O.H.); nasr.gomaa@science.bsu.edu.eg (N.H.G.); 2Department of Botany and Microbiology, Faculty of Science, Cairo University, Giza 12613, Egypt; wamer@sci.cu.edu.eg; 3Department of Biology, College of Science, Princess Nourah bint Abdulrahman University, P.O. Box 84428, Riyadh 11671, Saudi Arabia

**Keywords:** *Ephedra* species, *Ephedra* genders, fatty acids, lipid content, taxonomy, Gymnospermae, Egyptian flora

## Abstract

The genus *Ephedra* Tourn. ex L. (Ephedraceae) still exhibits taxonomic complexity that has not yet been resolved. This study aimed to determine the taxonomic identity of the *Ephedra* species in Egypt and identify the fatty acid profile and its diversity at the gender level as a taxonomic tool for specimens lacking reproductive cones. The current study provides a pioneering approach that distinguishes *Ephedra* species at the gender level. A total of 120 fresh individuals were collected from 20 locations representing different habitats where *Ephedra* plants grow in Egypt. In addition, herbarium specimens were deposited in Egyptian herbaria. The studied morphological traits included 30 vegetative characteristics and 72 traits of the reproductive organs of both genders. The fatty acid content was measured using gas–mass chromatography (GC-Mass). The taxonomic revision revealed that the Genus *Ephedra* was represented in the Egyptian flora by five species, *Ephedra alata* in section Alatae and *E. aphylla*, *E. ciliata*, *E. foemina*, and *E. pachyclada* in section *Ephedra*. South Sinai hosts these five species and represents the center of diversity for this genus in Egypt. The vegetative characteristics were subjected to principal component analysis (PCA), which revealed a distinct separation of the five studied species. Similarly, the cone traits treated by hierarchical clustering revealed intra-gender variations. The taxonomic key was developed based on the morphological traits to distinguish the studied species at the gender level. In total, 51 fatty acids were identified from the studied species and grouped as 18 saturated, 16 monounsaturated, and 17 polyunsaturated fatty acids. In the absence of reproductive cones, the lipid content and fatty acid composition of the vegetative parts displayed significant interspecific and intra-gender variations. Therefore, fatty acids can be used to efficiently identify the studied species when they lack reproductive cones. This study proved the efficacy of a multidisciplinary approach to identify *Ephedra* species at the gender level and recommends this trend for future studies of this genus.

## 1. Introduction

Globally, Gymnospermae include 873 species in 14 families [1]. Gymnospermae are represented in the Egyptian flora by two families, Cupressaceae and Ephedraceae [2]. Ephedraceae is a monogeneric family represented by a single genus, *Ephedra* Tourn. ex L. Worldwide, this genus includes 68 species [3,4], or 69 species [5], or 73 accepted species (https://powo.science.kew.org/ accessed on 1 July 2024). It is distributed across both the New World and Old World [6,7]. *Ephedra* species are typically perennial dioecious shrubs or under-shrubs [5] with xeromorphic features, assimilating branches, and opposing or whorled leaves that are frequently reduced to membranous sheaths [4,8]. The flowering of *Ephedra* species occurs between February and March. They have flowers in small cones: male flowers are subtended by a bract, two-lipped perianth, and staminal column with 2–9 anthers; female flowers are solitary or in groups of 2–3, subtended by 2–4–(6) pairs of bracts; and ovules have scarious or fleshy bracts.

Amongst Gymnosperms, the genus *Ephedra* is remarkably notorious for its research scarceness [9]. Taxonomically, it is classified into three sections based on the female cone bracts, which appear thickened, fleshy, and colorful in section Ephedra, dry–membranous, free, and winged in section Alatae Stapf, and free and coriaceous–dry in section Asarca Stapf [10].

The genus *Ephedra* is represented in Egyptian flora by two sections, Alatae and Ephedra. Section Alatae is represented by *E. alata* (tribe Tropidolepides). Section Ephedra includes two tribes, namely, tribe Scandentes, which includes three species (*E. aphylla*, *E. foliata*, and *E. foemina*), and tribe Pachyclada, which includes *E. pachyclada* subsp. *sinaica* [4,11]. Out of these five species, *E. alata* and *E. pachyclada* are among the “Least Concern Species″ according to the IUCN.

Although the genus *Ephedra* possesses a limited number of species, its species exhibit taxonomic complexity that has not yet been resolved [1]. The genus *Ephedra* has been subjected to earlier taxonomic and systematic studies [6,12,13,14]; in Egypt, it has been subjected to only floristic surveys [8,15,16]. For example, Täckholm [15] reported four *Ephedra* species, namely, *E. alata* Decne., *E. aphylla* Forssk., *E. ciliata* Fischer, and *E. campylopoda* C. A. Mey. Meanwhile, Boulos [16] added the *E. pachyclada* Boiss. to the aforementioned species. Later, Faried et al. [4] documented five *Ephedra* species: *E. alata* Decne., *E. aphylla* Forssk., *E. foeminea* Forssk., *E. foliata* Boiss., and *E. pachyclada* Boiss. subsp. *sinaica* (Riedl). However, their study did not cover a description of *E. foeminea* or the genus’s geographic range. Research has reported that the taxonomy of species and their gender identity are highly complex due to the lack of reproductive organs [4], the scarcity of specimens with intact mature reproductive organs, and the unclear morphological traits in stems and leaves [6,17].

The fatty acid composition of Gymnospermae can be used as a powerful taxonomic tool [1]. For example, the 5-olefinic acid is a characteristic fatty acid of Gymnospermae that Angiospermae does not synthesize, except in the family Rananculaceae [1]. Fatty acids from the leaves of 50 Gymnospermae species belonging to 14 families were studied earlier [1,18,19,20]. The later authors studied the fatty acids of the photosynthetic parts of 137 species belonging to 14 families, including five *Ephedra* species. Unfortunately, none of them was a Mediterranean species.

Most studies that have investigated the fatty acid composition in Gymnospermae do not specify the gender level (male and female individuals), and such data are still lacking. There are very scarce investigations into the fatty acid profile at the gender level in Gymnospermae. Hierro et al. [21] investigated the fatty acids of male and female *Ginkgo biloba*, but there is no sufficient work on Gymnosperms that can be generalized. Concerning the genus *Ephedra*, Nokhsorov et al. [22] investigated the role of lipids in *Ephedra monosperma* as an adaptive response to low temperature, but they did not make any reference to the gender level. Despite the importance of the fatty acid composition as an adaptation strategy, the lipid profile of *Ephedra* species at the gender level is still lacking.

During our field observations, we noticed confusion in the identification of the different *Ephedra* species because *Ephedra*′s morphological characteristics were reduced, and there are only a few characters, so the taxonomy of the genus *Ephedra* Tourn. ex L. has always been dubious and has been partially studied. Because the majority of the genus′s categorization has been based on restricted vegetative features, including the leaf length, female cone bracts, number of seeds per female cone, and plant habit, the infrageneric relationships among *Ephedra* have remained unclear.

There is still a gap in the previous studies that enables taxonomists to distinguish between the genders in specimens lacking reproductive cones. Fortunately, the lipid profile and fatty acid profile may also play a role in discriminating the different *Ephedra* species and may differ between both genders of the same species. Therefore, fatty acids and the lipid content could be an efficient tool in the identification of species when specimens lack reproductive organs. This study was applied to the Egyptian *Ephedra* species and aimed to (1) use morphological traits for both species and gender identification, (2) apply the fatty acid composition for both species and gender identification, and (3) use the fatty acid composition to differentiate between male and female specimens.

## 2. Results and Discussion

### 2.1. Taxonomic Investigation

#### 2.1.1. *Ephedra alata* Decne. (Section Alatae) Ann. Sci. Nat., Bot. sér. 2, 2: 239 (1834)

Syn. *E. alata* var. decaisnei Stapf, Denkschr. Kaiserl. Akad. Wiss. Wien. Math.-Naturwiss. Kl. 56(2): t. 1/1 (1889) nom. Inval.

Erect shrub up to 1.0 m tall. Stem circular, moderately branched at base. Branches grooved, papillose surface, stout, opposite, and scabrous. Leaves elliptical, 1.8–3 × 0.8–1.5 mm, opposite, sparsely ciliate, united for more than half their length (Figure 1). Cone bracts broad and woody with free scarious margins. Female cones grouped in axillary clusters carried on the straight peduncle. Cone bracts 4–5(–6) free pairs, each less than 1/3 cone length, orbicular, rounded apex, shortly lacerate margin. Outer bracts of strobilus 4–5 × 2–3 mm, equal in length. Inner bracts of strobilus 6–7.5 × 3.5–4 mm, less than twice as long as next bract. Fruit dry, creamy, cone-like, carried on erect peduncle. Seeds 6.5–8 × 2.5–4 mm, (1)–2 ovules/cone, flattened ovoid, brownish, 1–2 mm longer than inner female bract; micropylar tubule straight, 2–3 mm long (Figure 2 and Appendix A). Male cones grouped in axillary clusters, bracts 11–17 free pairs, each 2.5–3.2 × 1.2–1.5 mm, elliptical–obovoid, entire, acute–rounded apex, sub-equal. Strobilus bracts 6–12 pairs, each 1.5–3 × 1–2 mm, obovate, entire, retuse–rounded apex, obtuse–rounded base, sub-equal. Antherophore 2.5–5 mm branched above, anthers 3–6/strobilus, umbellate form (Figure 3 and Appendix A).

#### 2.1.2. *Ephedra aphylla* Forssk. (Section Ephedra), Fl. Aegypt.-Arab.: 170 (1775)

Syn: *E. alte* C. A. Mey., Monogr. Ephedra, Mém. Acad. Sci. Pétersb. 5: 75 (1846).

Climbing shrub up to 1.5 m tall. Stem circular, moderately branched at base. Branches moderately grooved, papillose–ciliate surface, tortuous, whorled, and scabrous. Leaves triangular, 2–4 × 0.4–0.8 mm, opposite-whorled, densely ciliate, united for less than half their length (Figure 1). Cone bracts soft, and green, later transformed into red. Female cones grouped in dense axillary clusters carried on curved peduncles. Cone bracts 2–3 fused pairs, each less than 1/3 cone length, ovate, ciliate, acute–obtuse apex. Outer bracts of strobilus 0.8–3 × 1–2.5 mm, equal in length. Inner bracts of strobilus 1.5–5.5 × 1–3 mm, almost equal to twice the length of the next bract. Fruit fleshy, red, berry-like carried on erect peduncle. Seeds 5–6.5 × 2.5–4 mm, 1–2 seeds/cone, ovate, brownish black, almost 1 mm longer than inner female bract; micropylar tubule straight, 0.4–0.6 mm long (Figure 2 and Appendix A). Male cones grouped in axillary clusters, bracts 11–15 fused pairs, each 1–3 × 0.8–1.2 mm, oblance–ovate, entire, rounded–obtuse apex, sub-equal. Bract of strobilus 7–10 pairs, each 1–3 × 0.5–1.5 mm, oblance–ovate, entire, rounded–obtuse apex, obtuse base, equal in length. Antherophore 2–5.5 mm unbranched above, anthers clustered 3–4/strobilus (Figure 3 and Appendix A).

#### 2.1.3. *Ephedra ciliata* Fisch. & C.A.Mey. (Section Ephedra), Bull. Cl. Phys.-Math. Acad. Imp. Sci. Saint-Pétersbourg 5: 36 (1845)

Syn: *E. foliata* Boiss. ex C.A.Mey. Mém. Acad. Imp. Sci. Saint- Pétersbourg, Sér. 6, Sci. Math., Seconde Pt. Sci. Mat. 7(2): 297 (1846).

Climbing shrub up to 3 m tall. Stem circular, moderately branched at base. Branches with non-conspicuous grooves, ciliate surface, striated, flexuous and narrow, opposite-pseudowhorled. Leaves filiform, 5–15 × 0.1–0.5 mm, opposite-whorled, densely ciliate, united for less than 1/8 their length (Figure 1). Cone bracts soft, and green, later transformed into white or red. Female cones grouped in terminal clusters carried on the straight peduncle. Cone bracts 2–3 fused pairs, each less than 1/3 cone length, and ovate, ciliate, acute–obtuse apex. Outer bracts of strobilus 2–3 × 1–2 mm, equal in length. Inner bracts of strobilus 4.5–7 × 1.5–3 mm, more than twice as long as next bract. Fruit fleshy, white-red, berry-like carried on curved peduncles. Seeds 6–7 × 2.5–4 mm, 1–3 seeds/cone, elliptical, brownish black, almost equal to the length of inner female bract; micropylar tubule straight, about 0.5–0.8 mm long (Figure 2 and Appendix A). Male cones grouped in terminal clusters, bracts 11–15 pairs, each 1.5–2.5 × 0.9–1.5 mm, elliptical, entire, obtuse apex, sub-equal. Bract of strobilus 7–8 pairs, each 1.5–2.5 × 0.5–1.5 mm, oblance–ovate, entire, obtuse apex, acute–obtuse base, equal. Antherophore 1.5–2.5 mm unbranched above, anthers opposite 3/strobilus (Figure 3 and Appendix A).

#### 2.1.4. *Epherda foeminea* Forssk. (Section Ephedra), Fl. Aegypt.-Arab.: 219 (1775)

Syn: *E. campylopoda* C.A.Mey. Bull. Cl. Phys.-Math. Acad. Imp. Sci. Saint- Pétersbourg, 5: 34 (1845).

Climbing or prostrate shrub up to 4 m tall. Stem circular, branched all over. Branches non-conspicuous grooves, papillose surface, thin, wiry, opposite-whorled. Leaves elliptical 0.8–2 × 0.5–1.7 mm, opposite-whorled, glabrous, often reduced to a short membranous sheath (Figure 1). Cone bracts soft, and green, later transformed into red. Female cones grouped in dense axillary clusters carried on the curved peduncle. Cone bracts 3-4-(5) fused pairs, each less than 1/3 cone length, and elliptical, entire, obtuse apex. Outer bracts of strobilus 3–3.5 × 1–2 mm, sub-equal in length. Inner bracts of strobilus 5–7 × 2–3 mm, almost equal twice the length of the next bract. Fruit fleshy, red, berry-like, carried on curved peduncle. Seeds 6–8.5 × 1.8–2.5 mm, 1–2 seeds/cone, elliptical, brownish, 1–2 mm longer than inner female bract; micropylar tubule straight, 0.3–0.5 mm long (Figure 2 and Appendix A). Male cones grouped in dense axillary clusters, bracts 11–12 pairs, each 1.5–2 × 0.9–1 mm, elliptical, entire, rounded apex, sub-equal. Bracts of strobilus 7–8 pairs, each 2.5–3 × 1–1.5 mm, elliptical, entire, obtuse apex, obtuse base, sub-equal. Antherophore 2.5–4 mm unbranched above, anthers clustered 4–7 (Figure 3 and Appendix A). *E. foemina* is the only creeping and insect-pollinated species within the studied species [23,24].

#### 2.1.5. *Ephedra pachyclada* Boiss. (Section Ephedra), Fl. Orient. 5: 713 (1884)

Syn. *E. sinaica* Riedl, Notes Roy. Bot Gard. Edinb. 38: 291 (1980).

Erect small shrub up to 40 cm tall. Stem circular, densely branched at nodes. Branches with non-conspicuous grooves, smooth–minutely papillose surface, stiff, thickened upright, opposite. Leaves elliptical 1.8–2.5 × 1–1.5 mm, opposite, glabrous, often reduced to a short membranous sheath (Figure 1). Cone bracts soft, and green, later transformed into red. Female cones grouped in dense axillary clusters on the short straight peduncle. Cone bracts 2–3 fused pairs, each less than 1/3 cone length, and ovate, entire, acute apex. Outer bracts of the strobilus 3–5 × 2–3 mm, equal in length. Inner bracts of strobilus 4–6 × 2–3.5 mm, less than twice as long as the next bract. Fruit fleshy, red, berry-like on curved peduncle. Seeds 4–5.5 × 2.3–3 mm, 1–(2) seeds/cone, elliptical, brownish, 0.5–1 mm shorter than inner female bract; micropylar tubule curved, 1.5–3.5 mm long (Figure 2 and Appendix A). Male cones grouped in dense axillary clusters, bracts 6–9 pairs, each 1.5–2 × 0.8–1 mm, narrowly oblong, entire, rounded apex, sub-equal. Bracts of strobilus 4–6 pairs, each 1.5–2 × 1.5–2 mm, orbicular–circular, entire, rounded apex, rounded base, sub-equal. Antherophore 2–2.5 mm unbranched above, anthers whorled, 6–8 (Figure 3 and Appendix A).

### 2.2. Species and Gender Key Based on Morpho-Taxonomic Traits

This constructed key was based on the current investigation and is supported by earlier works, among them [4,15,16,25].

**I.** Margins of cone bracts and leaf sheath glabrous ……………………….………… II


**+** Margins of cone bracts and leaf sheath ciliated……...…………………………….III

**II.** Stem erect, much branched at nodes, branches whitish-green, stiff, thickened upright ………………………………………………………………...…………….. *E. pachyclada*
Seed 1–(2) ovules/cone, micropylar tubule curved………………………... FemaleAnthers whorled 6–8, sessile and unbranched above, 4–6 strobili/cone ….. Male


**+** Stem climbing and prostrate, branched all over, branches dark green, thin, wiry ………………………..……………………………………………………....*E. foemina*
Seed 1–2 ovules/cone, micropylar tubule straight ……………..…………...FemaleAnthers clustered 4–7, sessile and unbranched above, 7–8 strobili per cone...Male


**III.** Cone bracts free with membranous edge……………………………….…... *E. alata*
Fruit creamy, cone-like, seed (1)–2 ovules/cone…………………...………...... FemaleAnthers umbellate 3–6, stalked and branched above, 7–12 strobili/cone....… Male


**+** Cone bract fused without membranous edges…………………………………...IV

**IV.** Origin of female cone terminal……………………………………………... *E. ciliata*
Fruit white-red, berry-like, seed 1–3 ovules/cone………………………..…FemaleAnthers opposite 3, sessile and unbranched above, 7–10 strobili/cone …......Male


**+** Origin of female cone axillary………………………….…………….….....*E. aphylla*
Fruit red, seed 1–2 ovules/cone………………………….………….………FemaleAnthers clustered 3–4, sessile and unbranched above, 7–10 strobili/cone…….Male


### 2.3. Morphometric Diversity at Interspecific and Intra-Generic Levels

Faried et al. and Price [4,13] mentioned that the nomenclature and history of the Mediterranean *Ephedra* species are chiefly complicated. Moreover, *Ephedra alte* C. Meyer of P. Forsskål was identified later as female *E. foliata* Boiss ex Meyer and male *E. aphylla*; this confusion was enhanced by the close morphology, ecology, and distribution range [26,27]. The current investigation of the five Mediterranean *Ephedra* species revealed several distinguishable features. The leaf length in the studied *Ephedra* species is notably variable among species as well as between the genders of each species. The total leaf length and the length of the free part of *E. ciliata* female and male individuals were notably longer than those in the other species, while the length of the united part of the leaf and leaf sheath of *E. alata* was longer than in the other individuals (Figure 1). Leaf sheath and cone bracts are glabrous in *E. foemina* and *E. pachyclada*, while both are ciliated in *E. alata*, *E. aphylla*, and *E. ciliata.* These data were reported earlier by Danin and Hedge and Faried et al. [4,25]. *E. foemina* is the only creeping and insect-pollinated species, while the rest are wind-pollinated and erect species [24].

*Ephedra alata* (section Alatae) is easily distinguished from other *Ephedra* species by its smooth stem surface; dry, membranous, wavy, winged female cone bracts; two seeds per cone (Figure 2); and a branched antherophore, which is branched in the other four species (Figure 3). These features are congruent with the earlier references [4,8,16,28,29,30,31,32,33,34]. Comparable to the section Ephedra, which includes the other four species (*E. aphylla*, *E. ciliata*, *E. foemina*, and *E. pachyclada*), this section is differentiated by fleshy and un-winged female cone bracts (Figure 2).

Figure 4 shows the PCA based on 30 vegetative characteristics of the studied species, at dimensions 1 and 2, with a cumulative variance of 49.2% for dimension 1, declaring the clear separation of the studied species into two groups along this dimension: the right group with three species, *E. alata*, *E. aphylla*, and *E. ciliata*, and the left group representing *E. foemina* and *E. pachyclada*. This grouping depends mainly on the absence of trichomes in the species of the left group. The higher width of the node up to 5 mm and clear stem grooves distinguish *E. alata*, the triangular shape of the leaf distinguishes *E. aphylla*, while the longer leaf length up to 15 mm and the high density of trichomes distinguish *E. ciliata*. While *E. foemina* is distinguished from the species of the right group by its climbing and prostrate habit as well as its branching all over the stem, *E. pachyclada* is distinguished from other species by the absence of a free part of the leaf and the dense branches at the node (up to 13/node). The achieved PCA grouping is analogous to the taxonomic identity of these species reported in previous studies [4,8,16].

Figure 5 outlines the hierarchical clustering analysis based on the 72 morphological characteristics of the cones in the studied species, revealing the separation of each gender (male or female) into a separate group. The main separating characteristics for the male gender were the numbers of both male cones and anthers/strobili, while the length of both the female cones and the inner bracts were the efficient traits in the female gender grouping (Figure 6). The characteristic of staminate cones in *Ephedra* is a distinctive feature, especially for the *E. alata* characterized by branched antherophore and stalked anthers, while other species are characterized by unbranched antherophore and sessile anthers [29].

The bract length of the female cone is not only a distinctive characteristic (Figure 6), but its size also showed more significant dissimilarity between the studied species than the male cone (Figure 6). The bract size of the female cone in female *E. alata* was significantly longer than that in the other individuals. The female cone traits are also the main distinctive characteristics for the Chinese *Ephedra* species [10].

### 2.4. Species Distribution and Habitats in Egypt

*Ephedra alata*, *E. aphylla*, and *E. ciliata* are distributed in the Egyptian deserts, including Sinai, in addition to the Mediterranean strip; *E. ciliata* is the least prevalent species. Each species possesses a characteristic habitat, desert sandy plains for *E. alata*, calcareous slopes and wadi beds for *E. aphylla*, while *E. ciliata* inhabits rocky slopes. *E. foemina* and *E. pachyclada* are very rare in prevalence with a restricted distribution range, as both grow hanging on rocky cliffs. *E. foemina* was traced in a few individuals in Sinai and the Mediterranean strip, while *E. pachyclada* was restricted to South Sinai. The distribution range and prevalence status recorded in our study were in harmony with the earlier floristic surveys [2,4,15,16]. Individuals were collected from 20 localities (indicated by stars in Table 1).

### 2.5. Lipid Content and Fatty Acid Composition at Interspecific and Intra-Generic Levels

The lipid content of the assimilating parts demonstrates significant interspecific variations. Figure 7 shows that the greater total lipid content was recorded in *E. ciliata* males (66 mg/g), followed by *E. alata* females (64 mg/g), while the lowest concentrations were reported in the female of *E. foemina* and *E. ciliata* (30 and 39 mg/g, respectively). Moreover, the intra-gender variation was notable for each species; nevertheless, the genders of each species were collected from the same population. Appendix A shows that the males of *E. ciliata*, *E. pachyclada*, and *E. foemina* have greater total lipid contents (66, 56, and 51 mg/g, respectively) than the female gender. By contrast, male *E. alata* and *E. aphylla* contained lower lipid contents (45 and 40 mg/g, respectively) than the female gender.

The alternation in lipid content of the cell membrane is the chief response to environmental stresses [22]. Accordingly, the detected higher lipid content of the studied genders (male *E. ciliata* and female *E. alata*; Figure 7) may reflect their better response to environmental stresses than other genders. High lipid contents help plants overcome climatic stresses, such as drought and higher temperatures [35,36].

### 2.6. Fatty Acid Composition at Interspecific and Intra-Gender Levels

The heat map (Figure 8) presents the common fatty acids in the studied *Ephedra* species visualized by color intensity into three clusters. The first cluster grouped male individuals of *E. alata*, *E. aphylla*, and *E. foemina* in addition to female *E. aphylla* based on their higher percentages of palmitic, oleic, and linoleic acids. The second cluster grouped the females of *E. pachyclada* and *E. alata* with higher percentages of linoleic and rumanic acids, followed by palmitic acid. The third cluster comprised male *E. pachyclada* and female *E. foemina* together with female and male *E. ciliata*, all containing high palmitic acid contents, and the latter species had high contents of oleic and myristic acids.

The studied *Ephedra* species collected from arid habitats showed the dominance of saturated fatty acids palmitic (C16:0), myristic (C14:0), and stearic (C18:0) in descending concentrations (Figure 8 and Appendix A). These results align with those by the authors of [1,37], who claimed the dominance of palmitic acid in photosynthetic tissues of the Gymnospermae.

The detected fatty acid composition revealed that fatty acids can distinguish between genders (male and female) of the same species, where valeric, cervonic, timnodonic, myristoleic, pentadecenoic, arachidonic, and docosatetraenoic acids are among the fatty acids present in the female gender only of *E. alata*, *E. aphylla*, *E. ciliata*, *E. foemina*, and *E. pachyclada* (Appendix A). Moreover, male *E. ciliata* and *E. pachyclada* lack polyunsaturated fatty acids, which could be used to distinguish males from females in the absence of reproductive organs. The absence of polyunsaturated fatty acids in the males of both species may induce their fragility to environmental stresses [35,36]. Our results show that fatty acids in Gymnospermae are not only a powerful taxonomic tool [1] but may extend to distinguish the gender of the species in the genus *Ephedra*.

The identified 51 fatty acids are grouped into 18 saturated (C4:0: C30:0), 16 monounsaturated (C14:1: C20:1), and 17 polyunsaturated (C16:2: C22:6) fatty acids. The radar plot (Figure 9) demonstrates the relative percentages of the saturated, monounsaturated, and polyunsaturated fatty acids in the studied *Ephedra* genders. Male *E. ciliata* possess the highest percentage (79.9%) of saturated fatty acids, followed by female *E. foemina* (75.46%), while the lowest percentages were recorded in female and male *E. alata* (31.99 and 37.23%, respectively). Male *E. foemina* possesses the highest percentage (50.03%) of monounsaturated fatty acids, followed by male *E. alata* (48.30%), and the lowest percentage is reported for female *E. alata* (5.15%). *E. alata* females have the highest percentage (62.86%) of polyunsaturated fatty acids, followed by *E. alata* males (48.30%), and the lowest percentage is recorded for females of *E. alata* (5.15%). Males of *E. ciliata* and *E. pachyclada* are characterized by their lack of polyunsaturated fatty acids. Nokhsorov et al. [22] reported that, in *E. monosperma* during an extreme reduction in temperature, the concentrations of unsaturated fatty acids (C18:2 and C18:3) increased while the saturated fatty acids, such as C16:0, decreased.

The female gender of the studied *Ephedra* species showed higher percentages of polyunsaturated fatty acids (4.3 and 62.8 in *E. foemina* and *E. alata*, respectively) than the male gender (Figure 9 and Appendix A). This may enable the female gender to adapt better to climatic changes [36].

The male gender had greater percentages of monounsaturated fatty acids than the female gender, except for *E. ciliata*, which showed a lower percentage (Figure 9 and Appendix A). However, the fatty acids of *Ephedra* species collected from cooler habitats showed the dominance (up to 50%) of the polyunsaturated fatty acids linolenic (C18:3) and linoleic (C18:2) in descending order [1]. The polyunsaturated fatty acids enhance the species′ adaptability to climatic changes [35,36]. In *E. monosperma*, the concentration of polyunsaturated lipid compounds of phospholipids and fatty acids increased during cold acclimatization [22].

At the date of issue, the fatty acid composition of Gymnospermae species to the gender level has not yet been fully uncovered, except by Hierro et al. [21], who differentiated between male and female Gymnospermae genders in *Ginkgo biloba* based on fatty acids. Our work is a pioneer in detecting the fatty acid composition in *Ephedra* species to the gender level and reveals that fatty acids possess highly distinctive concentrations and types at the intra-gender level. These findings are supported by Hierro et al. [21].

Pearson′s correlation analysis (Figure 10) was carried out to explore the relationship between the common fatty acids in the studied *Ephedra* species/genders. The analyses showed significant positive correlations between some fatty acid pairs: myristic–behenic, stearic–arachidic, and linoleic–rumenic. Significant negative correlations were also observed between margaric–lauric, palmitoleic–margaric, palmitic–linoleic, elaidic–rumenic, and vaccenic–linoleic (Figure 10).

## 3. Materials and Methods

### 3.1. Plant Material

A total of 120 fresh individuals were collected from 20 localities (indicated by stars (“*″) in Table 1) representing all the different habitats of *Ephedra* in two successive years (2023–2024). A total of 24 fresh specimens were examined for each species (12/each gender), collected from South Sinai, the Eastern desert, and Marsa Matruh in the western Mediterranean strip. *Ephedra* species were identified according to the contributions of earlier floristic and taxonomic treatments [4,15,16]. In addition to the fresh samples, herbarium specimens from Cairo University Herbarium (CAI), the Agricultural Museum (CAIM), the National Research Centre (CAIRC), the Desert Research Institute (CAIH), and Assiut University Herbarium (ASUT) were examined. Names and digital images and descriptions were examined as available in Jstor Global Plants (http://plants.jstor.org/ accessed on 1 January 2024), World Flora Online (https://www.worldfloraonline.org accessed on 30 January 2024), Kew herbarium (https://www.kew.org accessed on 13 March 2024), and the conifer online database (https://www.conifers.org accessed on 15 December 2023). The used names were adopted according to the International Plant Names Index and World Checklist of Vascular Plants 2024 (http://www.ipni.org & https://powo.science.kew.org accessed on 29 December 2023). Voucher specimens were deposited in the Cairo University Herbarium (CAI) and Beni-Suef Herbarium (BSH). For further investigations, fresh samples were stored in formol-acetic–alcohol (FAA: 50 mL ethyl alcohol, 10 mL formaldehyde, 5 mL glacial acetic acid, and 35 mL distilled water).

### 3.2. Morphological Investigations

A total of 24 fresh individuals and 30 strobili/species were morphologically investigated in each species. A total of 93 and 74 macro-morphological characteristics were used for female and male individuals, respectively (including shrub features, branches, stems, leaves, cone bracts, male and female strobili, fruits, and seeds) (Appendix A).

### 3.3. Determination of Crude Lipid

The crude lipid was extracted by methanol and chloroform from one gram of the dried powder plant material using a Soxhlet device for 8–10 h. Following the extraction, the solvent evaporated at low pressure; lipid material was desiccated and determined [38].

### 3.4. Identification of Fatty Acids Using Gas–Mass Chromatography (GC-Mass)

Fatty acids were investigated using a Trace GC1310-ISQ mass spectrometer (Thermo Scientific, Austin, TX, USA) with a direct capillary column, TG–5MS (30 m × 0.25 mm × 0.25 µm film thickness). The temperature of the column oven was adjusted at 50°C and then raised by 5 °C/min to 230 °C and kept for 2 min. Finally, it was raised by 30 °C/min to 290 °C and held for 2 min. The helium flow rate was 1 mL/min. After a three-minute solvent delay, diluted samples containing one microliter were injected. The temperatures of the injector and MS transfer line were maintained at 250 and 260 °C, respectively. Mass spectra were obtained at 70 eV ionization voltages through the m/z 40–1000 range. Fatty acids were identified by comparing the retention times and mass spectra with mass spectral databases from WILEY 09 and NIST 11 [39,40].

### 3.5. Statistical Analysis

The data were analyzed using the software package SPSS, version 20.0 (IBM Corporation, Armonk, NY, USA). The data were first examined for normality and homogeneity of variances using the Kolmogorov–Smirnoff and Levene′s tests, respectively. All data exhibit normality, and all statistical comparisons were carried out using one-way analysis of variance (ANOVA) followed by Tukey′s post hoc test. Next, the Graph Bad Prism software version 8.4.2 was used to draw histogram plots. R software version 4.3.2 (Vienna, Austria) was utilized and loaded with the necessary packages [41]. Principal component analysis (PCA) was employed to examine a dataset consisting of continuous variables after installing the “factoextra″ and “FactoMineR″ packages in R [42]. The correlation coefficients for the interaction between variables were obtained and displayed using the “Corrplot″ program [43]. White with a 0 denotes no association between the two variables, while blue with a 1 shows a strong positive correlation. A red score of −1 indicates a significant negative association. Hierarchical clustering analysis (heat map) was employed to examine continuous variables after installing the “pheatmap″ and “RColorBrewer″ packages in R [44]. Finally, Microsoft Excel 365 was used to draw radar and combo plots (statistical significance was defined as *p* < 0.05 and non-statistical significance as *p* > 0.05).

## 4. Conclusions

The current study revealed that the genus *Ephedra* in Egypt includes five species, in two sections: *Ephedra alata* in section Alatae and *E. aphylla*, *E. ciliata*, *E. foemina*, and *E. pachyclada* in section Ephedra. The traits of the reproductive cones are more efficient than the vegetative features as a taxonomic tool for identifying *Ephedra* species at the gender level. Also, the fatty acid composition is a helpful taxonomic tool in identifying *Ephedra* species when the specimens lack reproductive organs, since fatty acids showed interspecific and intra-generic variations in both type and composition. The study revealed that three species of *Ephedra*, namely, *Ephedra alata*, *E. aphylla*, and *E. ciliata* (in descending order), possess a wider distribution range and a common prevalence pattern, compared to the two rare species, *E. foemina* and *E. pachyclada*. This study pays attention to the need for dedicating more conservation efforts to protect the area hosting the genus diversity in South Sinai, Egypt.

## Figures and Tables

**Figure 1 plants-13-02442-f001:**
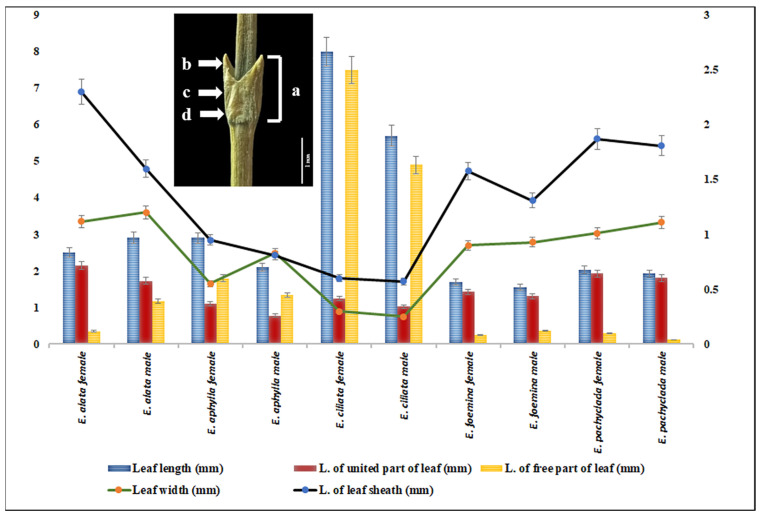
Combo plot of the leaf size (L × W mm): a: leaf length (mm); b: length of the free part; c: length of the united part of the leaf; and d: length of the leaf sheath of male and female *Ephedra* individuals.

**Figure 2 plants-13-02442-f002:**
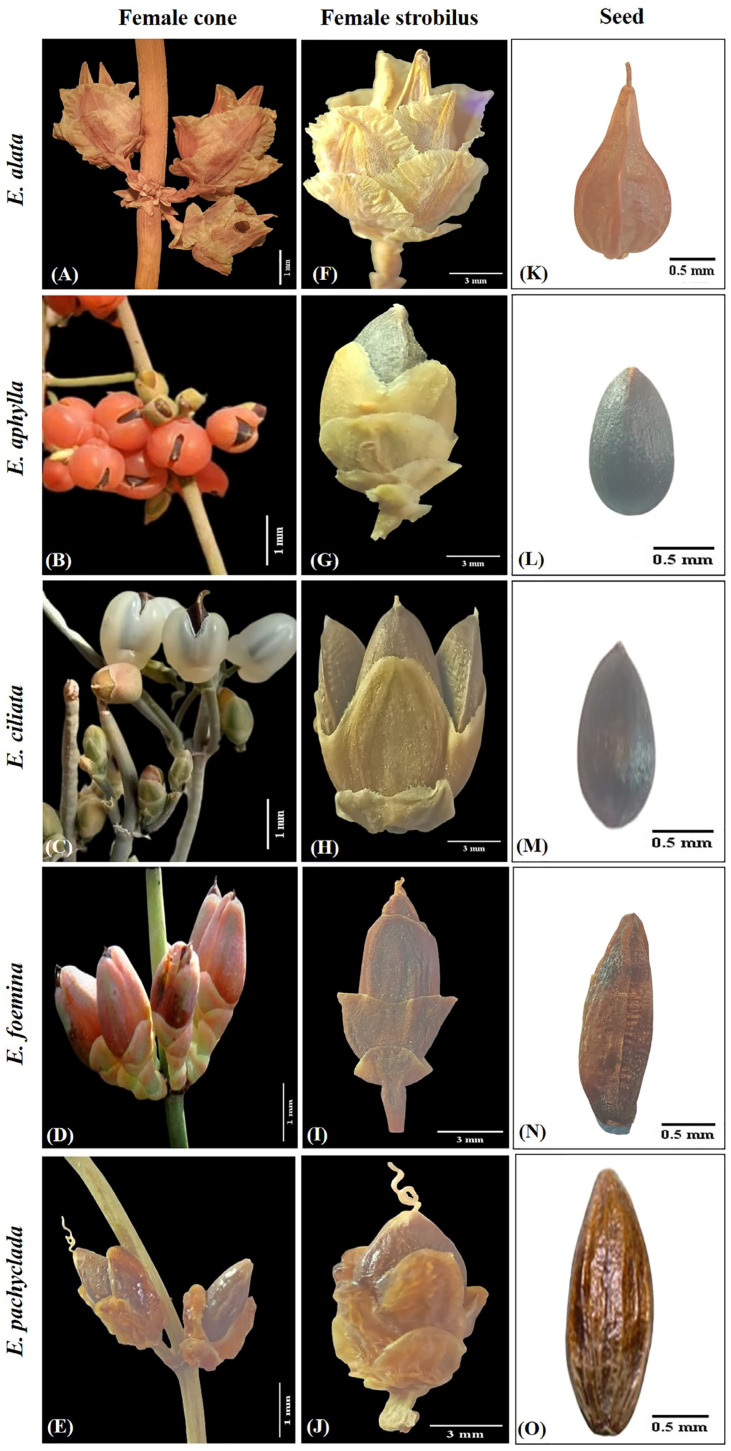
(**A**–**E**): female cones of the studied *Ephedra* species, (**F**–**J**): female strobilus of the studied *Ephedra* species, and (**K**–**O**): seeds of the studied *Ephedra* species.

**Figure 3 plants-13-02442-f003:**
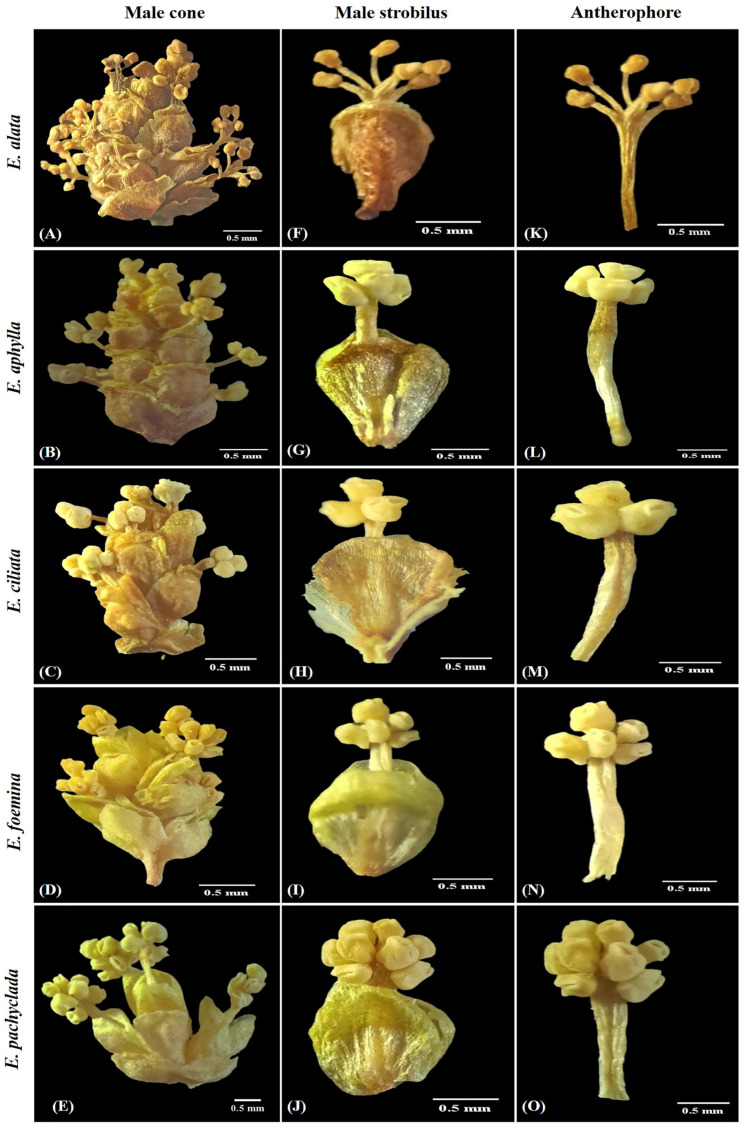
(**A**–**E**): male cones of the studied *Ephedra* species, (**F**–**J**): male strobilus of the studied *Ephedra* species, and (**K**–**O**): antherophore of the studied *Ephedra* species.

**Figure 4 plants-13-02442-f004:**
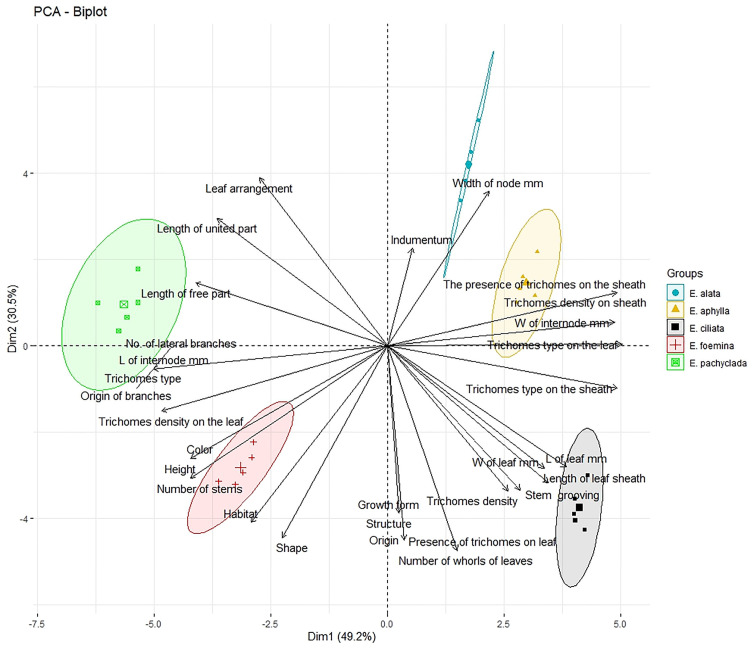
Principal component analysis (PCA) of the 30 morphological characteristics with 49.2% total variance described along the first axis (Dim 1) followed by 30.5% variance exhibited along the second axis (Dim 2).

**Figure 5 plants-13-02442-f005:**
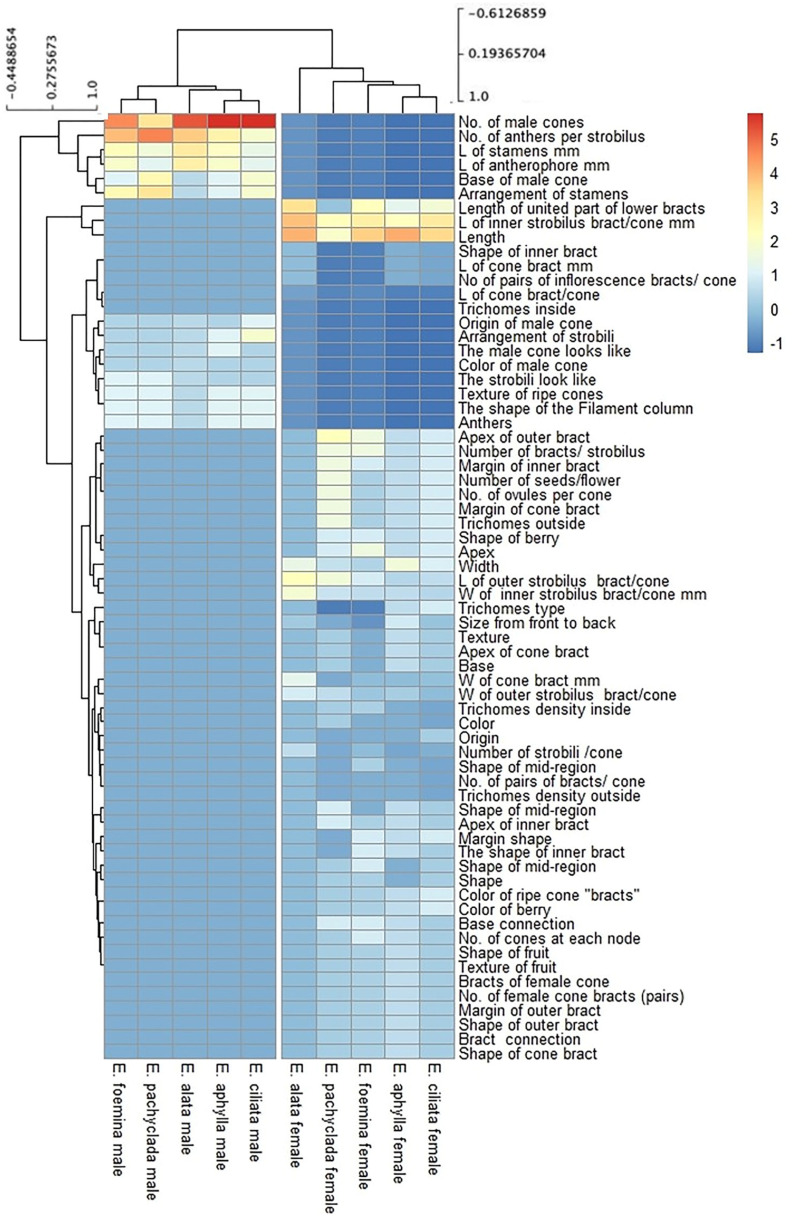
Hierarchical clustering analysis of morphological characteristics of male and female cones based on 72 characteristics. Data are represented by the means of at least 3 replicates.

**Figure 6 plants-13-02442-f006:**
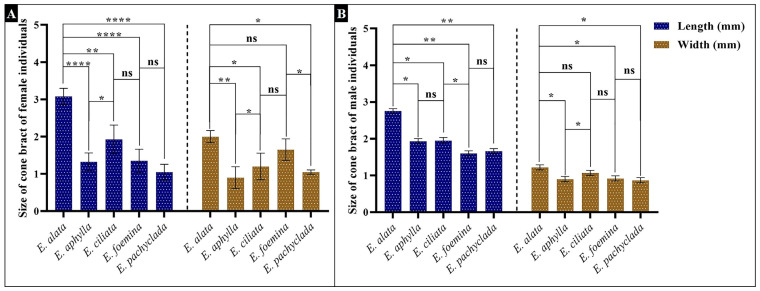
Size of cone bracts (L × W mm). (**A**) Female individuals, (**B**) male individuals. *p* value: * *p* ≤ 0.05, ** *p* ≤ 0.01, and **** *p* ≤ 0.0001. ns: not significant.

**Figure 7 plants-13-02442-f007:**
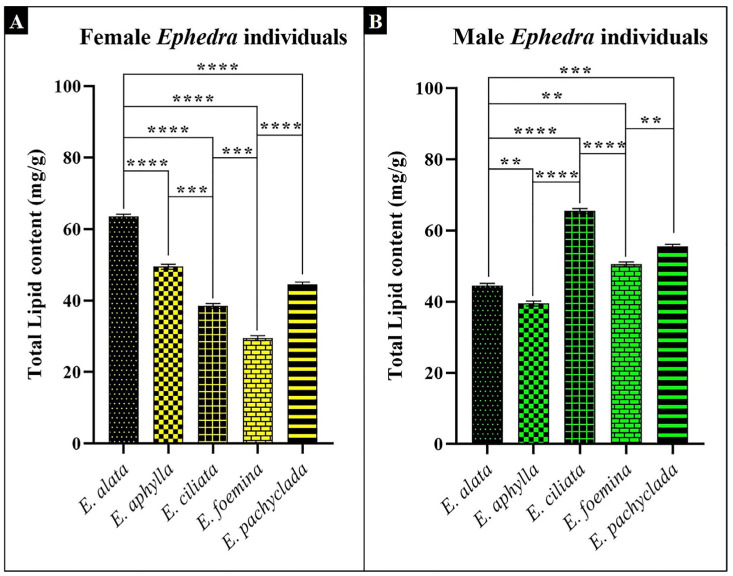
Concentration of the total lipid contents (mg/g) in the studied *Ephedra* species. (**A**) Female individuals, (**B**) male individuals. *p* value: ** *p* ≤ 0.01, *** *p* ≤ 0.001, and **** *p* ≤ 0.0001.

**Figure 8 plants-13-02442-f008:**
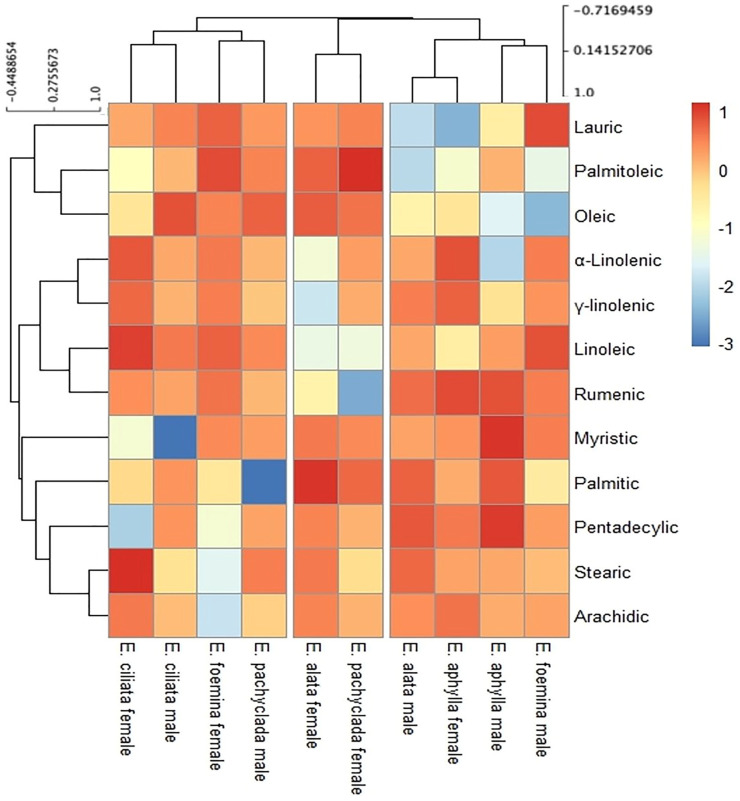
Hierarchical clustering analysis of most common fatty acids in *Ephedra* species. Data are represented by the means of at least 3 replicates.

**Figure 9 plants-13-02442-f009:**
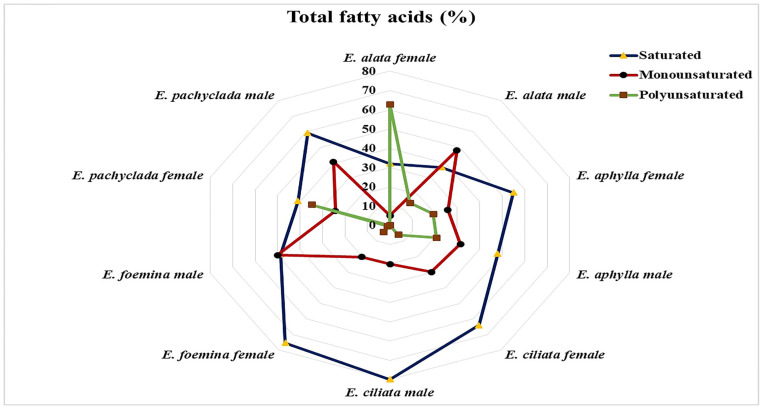
Radar plot of the fatty acids identified in the studied *Ephedra* species at the gender level.

**Figure 10 plants-13-02442-f010:**
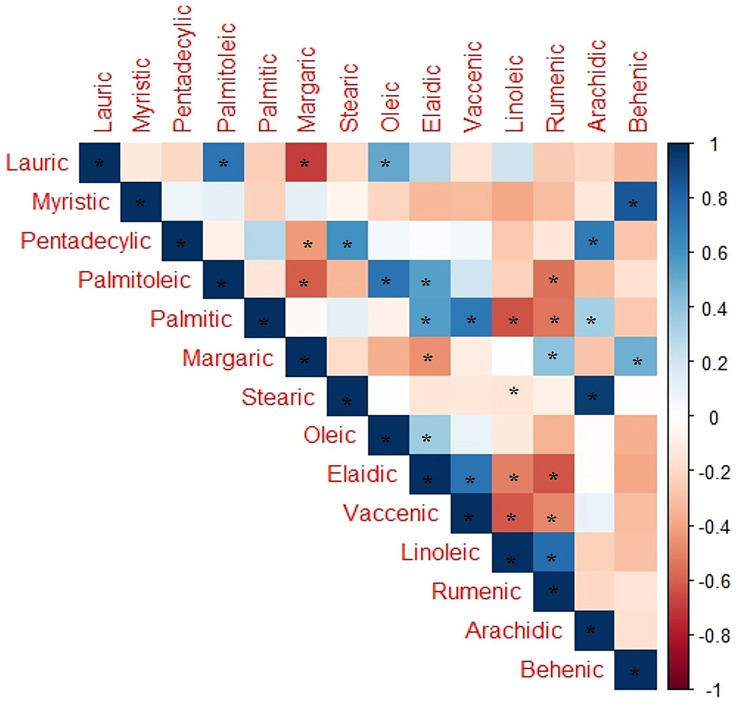
Pearson′s correlation analysis based on the correlation coefficient of the most common fatty acids in the studied *Ephedra* species. Positive and negative correlations are displayed in blue and red colors, respectively. Correlation coefficients are proportional to color intensity. (*) means strong positive and negative correlations.

**Table 1 plants-13-02442-t001:** Localities for the studied *Ephedra* species (localities are arranged from north to south).

Species	Locality	GPS Coordinates		Gender	Date
		N	E		
*E. alata*	Gebal El-Shaieb, Red sea	26°58′47″	33°29′11″	Male	11 February 2021
	Wadi Feiran, Sinai *	28°43′06″	33°37′06″	Female	15 May 2024
	S. Galala near Ras Zafarana	28°44′33″	32°23′17″	Male	6 February 1960
	Wadi Hamamet faraon, Sinai near red sea *	28°53′50″	33°22′22″	Female	17 June 2023
	Sinai: Abo Zeinema	29°02′31″	33°06′30″	Female and Male	14 June 2021
	Wadi Abu Khodirate, 85 Km from Zafarana	29°06′42″	32°39′35″	Female	14 March 1997
	Sinai: Wadi Fereeh	29°07′15″	33°10′47″	Female and Male	20 May 2021
	El Saff desert, Wadi Warag	29°20′54″	33°14′31″	Female	23 March 1982
	Hills of South Galala	29°21′29″	32°13′04″	Female	4 March 2021
	Wadi Araba, Eastern Desert *	29°34′39″	34°59′59″	Female	28 April 2023
	Wadi Houimela	29°36′27″	30°46′35″	Female	4 March 1973
	Wadi Hoff, Helwan desert *	29°52′51″	31°18′30″	Female and Male	14 March 2023
	Wadi At fihi, El Saff desert	29°56′54″	31°22′38″	Male	24 March 1982
	Gebl Katania, Suez road	29°58′00″	32°33′02″	Female	20 November 1998
	Wadi Katamiya, Suez road	30°00′25″	31°25′23″	Female and Male	11 March 1960
	Mitla Pass, Sinai *	30°00′41″	32°53′30″	Male	15 August 2023
	Cairo: Wadi Degla *	30°01′32″	31°19′09″	Female and Male	29 May 2023
	Wadi Gondall, Cairo, Suez Road	30°02′39″	31°14′08″	Male	9 April 1990
	Cairo, Suez, desert road	30°05′07″	31°57′20″	Male	15 March 2020
	Suez Road, kilo 20	30°05′09″	31°56′27″	Male	15 April 2008
	Suez Road, kilo 22	30°05′11″	31°56′14″	Female	15 April 2008
	Plateau between Qara and Marsa Matrouh	31°16′37″	27°11′37″	Male	5 April 1998
*E. aphylla*	W of Marsa Halaib, in the wadies	22°13′36″	36°38′44″	Female	22 January 1933
	Wadi Lethi, Saint Catherine, Sinai *	28°33′07″	33°58′24″	Male	15 July 2023
	Deir El-Rahba garden, Saint Catherin *	28°33′37″	33°56′52″	Female and Male	5 May 2024
	Sinai: Wadi El-Kid	28°34′47″	34°17′16″	Female and Male	28 March 2004
	Tabouk, Saint Catherine, Sinai at 1813 altitude *	28°54′08″	33°93′12″	Male	16 August 2023
	Wadi Al-Arbaeen Saint Catherine, Sinai at 1782 altitude *	28°55′88″	33°94′98″	Female and Male	17 August 2023
	W Qiseib, N. Galala	29°24′36″	32°28′38″	Male	9 February 1956
	Wadi Hoff, Helwan desert *	29°52′51″	31°18′30″	Female and Male	14 March 2023
	Wadi Bad of Red Sea Coast	29°41′39″	32°16′58″	Female	9 June 2021
	Cairo: Wadi Degla *	30°01′32″	31°19′09″	Female and Male	29 May 2023
	Cairo: Zohria garden	30°02′45″	31°13′35″	Female	15 November 1928
	Wadi El-Habes, before Agiba	30°02′48″	31°19′57″	Female and Male	23 March 1974
	Medicinal garden, Barrage	30°03′56″	31°08′38″	Female and Male	21 May 1979
	Gebel Elba	30°06′00″	31°20′05″	Female	21 February 1998
	Between Khanka and Abu Zaabal	30°13′53″	31°21′41″	Female	29 February 1960
	Burg El Arab: Bramly′s grotto *	30°52′25″	29°33′36″	Female and Male	15 July 2023
	Burg El Arab (Roman Cistern)	30°54′03″	29°35′05″	Male	11 March 2022
	On the coastal road 46 Km before Marsa Matrouh	30°59′24″	29°39′40″	Male	3 May 1998
	Mariout	31°09′04″	29°54′06″	Female	25 May 2008
	Ras El Hekma	31°10′17″	27°49′43″	Female	25 May 1997
	Alexandria: Vectoria	31°14′11″	29°58′10″	Female and Male	25 August 1921
	Khan Younis	31°20′46″	34°18′14″	Female	16 September 1955
*E. ciliata*	Marsa Halaib	22°14′00″	36°39′00″	Female	22 January 1929
	Gebl Hamata, Red Sea coast	24°20′21″	35°12′06″	Female	7 June 2009
	Wadi Marsa Kwan	26°33′16″	34°02′04″	Female	10 February 1962
	Sinai: Wadi Alletehi	28°09′73″	34°04′54″	Female and Male	11 April 2004
	Sinai: Wadi Al Ratam	28°23′90″	34°23′85″	Female and Male	28 March 2004
	Sinai: Wadi Gebal	28°32′28″	33°52′53″	Female and Male	28 April 2004
	Wadi Lethi, Saint Catherine, Sinai *	28°33′07″	33°58′24″	Male	15 July 2023
	Sinia, the garden of. St, Catherine′s Monastery *	″28°33′25″	33°58′23	Male	25 August 2023
	The Garden of Deir el Arba′ in Sinai *	″28°47′25″	33°35′23	Male	17 July 2023
	Al-Rassis Cathrine, Sinia at 1580 altitude *	28°55′88″	33°94′98″	Female and Male	15 May 2023
	Sinai: Wadi Reem	28°66′80″	33°66′74″	Female and Male	23 April 2004
	Sinai: Wadi Aleyaat	28°66′86″	33°65′37″	Female and Male	22 April 2004
	Al fred Bircher′s garden, El Saff	29°34′34″	31°17′28″	Female	22 August 2005
	Wadi Alkwamtra	29°38′12″	32°35′02″	Male	27 February 1967
	Wadi Yahameib	29°50′19″	31°08′59″	Female	5 April 2020
	Wadi Degla	30°01′32″	31°19′09″	Female and Male	29 May 2022
	Cairo: Zohria garden	30°02′45″	31°13′35″	Female	15 November 1928
	Gebel Elba	30°06′00″	31°20′05″	Female	21 February 1998
	Zaafaran Garden, Abbassia, Cairo	30°17′59″	31°29′39″	Male	16 June 2022
	Wadi El Habs, between Marsa Matruh and Agiba *	30°39′36″	29°21′57″	Male	7 August 2023
	El Mansouryeh	31°46′48″	35°14′42″	Male	23 May 1997
*E. foemina*	Gabal Mousa, Sinai *	28°54′13″	33°97′55″	Female and Male	16 July 2023
	Wadi Halazeen, 45 Km west of Marsa Matruh	31°11′27″	27°38′01″	Female and Male	25 April 2022
	Busaili, old Rashid Road, and 15 km before Rashid, Alexandria *	31°11′39″	29°56′40″	Female and Male	19 March 2023
	Brance, Marsa Matruh *	31°16′37″	27°15′49″	Female and Male	28 May 2023
*E. pachyclada*	Sinai: Wadi Al-Talaa, Al-Kabera	28°23′45″	33°52′45″	Female and Male	27 March 2004
	Sinai: Ain Al-Tofaha	28°32′54″	33°56′26″	Female and Male	28 March 2004
	Sinai: Wadi El-Kid	28°34′47″	34°17′16″	Female and Male	14 March 2004
	Abu Walia, Saint Catherine, Sinai at 1905 altitude *	28°53′55″	33°91′39″	Female	16 August 2023
	Abu Walia, Saint Catherine, Sinai at 1891 altitude *	28°53′60″	33°90′92″	Male	16 August 2023
	Gabal Mousa, Sinai at 2275 altitude *	28°53′83″	33°97′50″	Female and Male	27 August 2023
	Gabal Mousa, Sinai at 2260 altitude *	28°53′92″	33°97′53″	Female and Male	27 May 2024

*: collected samples.

## Data Availability

Data are contained within the article.

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
