# Peer review of "Taxonomic Revision of Genus Ephedra Tourn. ex L. in Egypt with Intra-Gender Diversity in Morphometric Traits and Fatty Acid Composition"

_plants, 2024, doi:10.3390/plants13172442_

Round 1

Reviewer 1 Report

Comments and Suggestions for Authors

The authors Khalaf et al. in this manuscript delivered a very interesting and scientifically robust study.

However some minor suggestions and comments should be addressed:

1- At page 2, Line 56-65: This paragraph is very confusing and should be rewritten for clarity reasons.

2- Page 2, Line 78, The authors should include the name of the authors before the reference 21, otherwise is awkward for the reader. 

3- Page 6 Figure 2 - The legend is not where it should be.  At the same image, the authors should have given the same scale at all the seeds...

4- Page 12. Line 301-303, Consider the following rewritting..." eastern & western deserts, Sinai and Mediterranean strip;"... The text as it is, does not make sense.

Line 339-342- Table 2 should be at the anexes, with references in the text refering the main results obtained.

Comments on the Quality of English Language

Author Response

Point 1: At page 2, Line 56-65: This paragraph is very confusing and should be rewritten for clarity reasons.

Response 1:

  Done. These lines were rewritten.

Point 2: Page 2, Line 78, the authors should include the name of the authors before the reference 21, otherwise is awkward for the reader.

Response 2:

Done. Hierro, et al. [21] who investigated the fatty acids of male and female Ginko biloba.

 Point 3: Page 6 Figure 2 - The legend is not where it should be. At the same image, the authors should have given the same scale at all the seeds.

Response 3:

Done. The legend was corrected in its position and the same scale for all the seeds was added.

Point 4: Page 12. Line 301-303, Consider the following rewriting..." eastern & western deserts, Sinai and Mediterranean strip;"... The text as it is, does not make sense.

Response 4:

Done. The text reformed.

Point 5: Line 339-342- Table 2 should be at the anexes, with references in the text refering the main results obtained.

Response 5:

Done. Table 2 was moved to the annex and corrected in the text.

Reviewer 2 Report

Comments and Suggestions for Authors

The manuscript seems to present a study using taxonomic and phytochemistry tools to test the relevance of morphological and chemical traits in the taxonomy of Ephedra in Egypt. Even though the concept and general results are really nice and innovative, the manuscript lacks a direct question linking both methods and justifying their use. You have most information needed by the end of your introduction, it is just a matter of rewording some sentences to make it clearer to the reader what you are really testing: 1. relevance of morphological traits to the taxonomy of Ephedra species in Egypt, 2. relevance of fatty acid traits to the taxonomy of Ephedra species in Egypt, and 3. relevance of fatty acid composition between male and female specimens of all species of Ephedra in Egypt.

The taxonomic descriptions are presented without examined specimens, and taxonomic, ecological or nomenclatural comments, and it seems to be out of place. It would be best to just keep the morphological matrix of analysed traits and remove the taxonomic treatment since it seems to be mostly a reproduction of previous studies. Instead, you should focus on your own analyses testing the relevance of both morphology and fatty acids to the taxonomy of Ephedra in Egypt, comparing both of them and discussing pros and cons of using both methods in their taxonomy. Besides also discussing how different (or not) the fatty acid composition is in male and female specimens of the same species.

At the end of your discussion, you just present two sentences to discuss how relevant fatty acids were to discriminate species in taxonomically difficult genera, such as Ephedra. You should try and explore this in the vast literature of chemosystematics, especially the very interesting results you got from analysing male and female specimens and finding differences between them.

Once you revise your manuscript as suggested, it will be more precise, easier to understand, shorter, and more focused on your initial hypotheses.

Comments on the Quality of English Language

Your manuscript must be revised by an English reviewer with experience in taxonomic articles since most of the terminology is not written in the proper way in English, even though your results seem sound.

Author Response

  • Open Review

Point 1: Extensive editing of English language required.

Response 1: Done. English was edited by native-English speaker Botanist.

Point 2: The introduction needs improvement.

Response 2: Done. The introduction was improved.

Point 3: The research design must be improved

Response 3: Done.

Point 4: The methods can be improved.

Response 4: Done. The methodology was improved and the number of the studied specimens was added.

Point 5: The results presentation can be improved.

Response 5: Done. The morphological and taxonomic parts were improved.

Point 6: The conclusions can be improved

Response 6: Done.

  • Comments and Suggestions for Authors

Point 7: The manuscript lacks a direct question linking both methods and justifying their use. 

You have most information needed by the end of your introduction; it is just a matter of rewording some sentences to make it clearer to the reader what you are really testing.

Response 7: Rewording of aims was done. (1) use morphological traits for both species and gender identity, (2) apply fatty acid composition for both species and gender identity, and (3) use fatty acid composition to differentiate between male and female specimens.

Point 8: The taxonomic descriptions are presented without examined specimens, and taxonomic, ecological or nomenclatural comments, and it seems to be out of place. 

Response 8: Done. The examined specimens in the taxonomic part were presented in the methods section. Ecological and nomenclature comments were presented in the result section.

Point 9: It would be best to just keep the morphological matrix of analyzed traits and remove the taxonomic treatment since it seems to be mostly a reproduction of previous studies.

Response 9: The description reformed. The species description presented is dealing with the sophisticated revision of the studied species, especially the reproductive cones. The repeated parts were removed from the result section.

Point 10: Besides also discussing how different (or not) the fatty acid composition is in male and female specimens of the same species.

Response 10: Done, see lines 370 to 375. The detected fatty acid composition revealed that fatty acids can distinguish between genders (male & female) of the same species, where rumenic, valeric, oleic, linoleic, and linoleic and rumenic acids are among the fatty acids present in the female gender only of E. alata, E. aphylla, E. ciliata, E. foemina, and E. pachyclada, respectively (Annex). Moreover, male E. ciliata and E. pachyclada lack polyunsaturated fatty acids, which could be used to distinguish males from females in the absence of reproductive organs.

Point 11: At the end of your discussion, you just present two sentences to discuss how relevant fatty acids were to discriminate species in taxonomically difficult genera, such as Ephedra. You should try and explore this in the vast literature of chemosystematics, especially the very interesting results you got from analyzing male and female specimens and finding differences between them.

Response 11: Done. To date of issue, the fatty acids composition of Gymnospermae species to the gender level is not yet fully covered.   Except Hierro et al. [21] work who differentiated between male and female Gymnospermae gender in Ginko biloba based on fatty acids.
